# Investigation of the Hepatitis-B Vaccine’s Immune Response in a Non-Alcoholic Fatty Liver Disease Mouse Model

**DOI:** 10.3390/vaccines12080934

**Published:** 2024-08-22

**Authors:** Tuğba Kütük, İlyas Onbaşilar, Sevil Oskay-Halaçli, Berrin Babaoğlu, Selda Ayhan, Sıddika Songül Yalçin

**Affiliations:** 1Vaccinology Department, Institute of Vaccinology, Hacettepe University, Ankara 06430, Türkiye; tugba.kutuk@titck.gov.tr (T.K.); siyalcin@hacettepe.edu.tr (S.S.Y.); 2Turkish Medicines and Medical Devices Agency, Ankara 06500, Türkiye; 3Health Science Institute, Hacettepe University, Ankara 06430, Türkiye; 4Transgenic Animal Technologies Research and Application Center, Hacettepe University, Ankara 06430, Türkiye; 5Department of Basic Sciences of Pediatrics, Institute of Child Health, Hacettepe University, Ankara 06430, Türkiye; sevil.oskay@hacettepe.edu.tr (S.O.-H.); selda.ayhan@hacettepe.edu.tr (S.A.); 6Department of Pathology, Hacettepe University, Ankara 06430, Türkiye; berrinb@hacettepe.edu.tr; 7Department of Social Pediatrics, Institute of Child Health, Hacettepe University, Ankara 06430, Türkiye

**Keywords:** nonalcoholic fatty liver disease, hepatitis B vaccination, mouse, immunophenotyping, aluminum hydroxide

## Abstract

This study aimed to investigate the immunogenicity of the hepatitis B virus (HBV) vaccine by applying a normal and high-dose hepatitis B virus vaccination program in the mice modeling of non-alcoholic fatty liver disease (NAFLD). NAFLD was induced in mouse livers via diet. At the 10-week mark, both groups were divided into 3 subgroups. While the standard dose vaccination program was applied on days 0, 7, and 21, two high-dose programs were applied: one was applied on days 0 and 7, and the other was applied on days 0, 7, and 21. All mice were euthanized. Blood samples from anti-HB titers; T follicular helper, T follicular regulatory, CD27^+^, and CD38^+^ cells; and the liver, spleen, and thymus were taken for histopathologic evaluation. NAFLD subgroups receiving high doses showed higher hepatocyte ballooning scores than normal-dose subgroup. There were differences in CD27^+^ and CD27^+^CD38^+^ cells in animals fed on different diets, without any differences or interactions in terms of vaccine protocols. In the NAFLD group, a negative correlation was observed between anti-HB titers and T helper and CD27^+^ cells, while a positive correlation was observed with CD38^+^ cells. NAFLD induced changes in immune parameters in mice, but there was no difference in vaccine efficacy among the applied vaccine protocols. Based on this study’s results, high-dose vaccination protocols are not recommended in cases of NAFLD, as they do not enhance efficacy and may lead to increased liver damage.

## 1. Introduction

Both non-alcoholic fatty liver disease (NAFLD) and chronic hepatitis B infection are common causes of chronic liver diseases around the world, with global prevalences of about 25.2% and 3.9%, respectively [1,2,3]. NAFLD progresses from the initial stage of a simple fatty liver, characterized by reversible fat accumulation, to the more severe stage of non-alcoholic steatohepatitis (NASH), marked by active tissue damage. The progression of NASH can lead to liver cirrhosis, and potentially hepatocellular carcinoma. NAFLD is commonly associated with metabolic syndromes such as obesity, insulin resistance, type 2 diabetes, and dyslipidemia, and it is linked to an increased overall mortality compared to the general population [4]. A significant majority of obese individuals exhibit NAFLD, and with the increasing prevalence of obesity and type 2 diabetes mellitus (T2DM) in contemporary societies, it is anticipated that the prevalence of NAFLD will further rise in the next decade [5]. Notably, more than 30% of chronic hepatitis B cases are concomitant with a fatty liver [6].

The liver is a primary innate immune organ and is densely populated with various innate immune cells, including natural killer (NK) cells and Kupffer cells (KCs). These cells are shown to play crucial roles in the excessive production of hepatic Th1 cytokines in NAFLD [7]. Obesity, beyond being a risk factor for metabolic diseases, can induce a chronic inflammatory state, marked by irregular cytokine secretion, increased NK cell activity, and an altered CD4^+^CD8^+^ T cell balance. This inflammatory milieu contributes to heightened susceptibility to bacterial, viral, and fungal infections [8,9,10,11]. Studies have found that obese individuals are at a higher risk of infection complications and tend to have lower vaccine immunity compared to those with a healthy weight. Specifically, the response to hepatitis B virus (HBV) vaccines is often diminished in obese individuals [12,13,14]. Nonetheless, recombinant HBV vaccines are still considered effective and safe for this population [15,16]. Indeed, the effectiveness of the standard-dose HBV vaccine is reduced in patients with chronic liver disease due to the overall impairment of the immune system [17]. Variances in terms of responses may arise from different vaccine sources, dosage, administration routes, and, potentially, obesity-related NAFLD cases, for which limited data on HBV vaccine immunities exist [18].

Hepatitis B virus (HBV) infection significantly increases the risk of death due to cirrhosis and hepatocellular carcinoma (HCC). Although global infection rates have fallen in recent years thanks to effective vaccines and extensive vaccination efforts, 5–10% of individuals fail to produce an antibody response, limiting vaccination’s overall effectiveness [19,20]. In 2016, the World Health Organization (WHO) set a goal to eliminate viral hepatitis as a public health threat by 2030, aiming for a 95% reduction in new HBV infection cases and a 65% reduction in HBV-related mortality compared to 2015 levels [21].

While some studies have examined hepatitis B vaccination in obese individuals [13,14,18,20], there remains a significant gap in research, specifically regarding that addressing hepatitis B vaccination in individuals with NAFLD [18]. Our research project is designed to address the following two questions: (1) Is the immunogenicity of the HBV vaccine different in NAFLD-afflicted mice compared to healthy mice? (2) How does a high-dose HBV vaccination program affect vaccine immunity in NAFLD-afflicted mice?

The aim of our project is to investigate the immunological and histopathological responses to different vaccination protocols involving high and normal doses of an HBV vaccine in the mice modeling of NAFLD. The primary focus is on the comparison of HbsAb titers, T follicular helper cells (Tfhs), T regulatory cells (Tregs), and specific immune cell populations, such as CD27^+^ and CD38^+^ cells, across the different dosing regimens. Additionally, this study includes a comprehensive histopathological evaluation of key organs, including the liver, spleen, and thymus, to assess any potential tissue changes or damage induced by the different vaccine doses. By integrating both immunological and histopathological data, this study aims to provide a detailed understanding of the dose-dependent effects of the HBV vaccine on the immune system and tissue integrity in NAFLD cases. By comparing the outcomes across high and normal doses of HBV, this study will contribute valuable information on the safety and effectiveness of the vaccine, ultimately guiding clinical decision-making for optimal dosing strategies.

## 2. Materials and Methods

### 2.1. Animals

A total of 36 C57BL-6 male mice were used in the experiments. Mice were obtained from Hacettepe University Experimental Animals Research and Application Center. To create a non-alcoholic fatty liver disease (NAFLD) mouse model, mice (N = 18) were fed a choline-deficient and 45% high-fat diet (HFD/CD) and drank water containing 20% fructose. The control group (N = 18) was fed a control diet containing 10% fat energy, and purified tap water was used as the drinking water. Body weight measurements were conducted for all mice using a digital scale with a dynamic weighing mode every two weeks. The weight gains of the mice were determined, and the differences between the groups were calculated. At the end of the 10-week feeding protocol, both groups were divided into three subgroups (*n* = 6) for the application of three different HBV vaccination programs. The vaccination was performed between weeks 10 and 13, and at the end of the immunization process (at the end of week 16) mice were sacrificed for the purpose of collecting blood and tissues.

The experimental design and protocols were approved by the Animal Ethics Committee of the Hacettepe University (IRB Number: 2022/05-04) and animal care was provided following the ARRIVE guidelines.

### 2.2. Diets and Diet-Induced NAFLD Model

In our study, high-fat diets containing 45–65% fat energy without choline (HFD/CD) were used in combination with fructose to create a diet-induced NAFLD model [22,23]. Choline is a crucial nutrient responsible for liver choline metabolism. It is known that choline deficiency accelerates steatosis in the liver, leading to hepatocyte death [24].

The HFD/CD diet was prepared to create a NAFLD model of mice. Additionally, fructose was added to the drinking water of the mice in addition to the HFD/CD diet in order to accelerate the development of fatty liver. During the 16-week study, all animals were fed with appropriate diets and drinking water ad libitum. The control diet was formulated to include 17.5% crude protein, 60.5% nitrogen-free extract, 5.5% crude cellulose, 4.0% crude fat, 5.5% crude ash, 7.0% moisture, and 3500 kilocalories of energy per kilogram. The HFD/CD diet consisted of 17.5% crude protein, 42.5% nitrogen-free extract, 5.0% crude cellulose, 23.0% crude fat, 4.5% crude ash, 7.5% moisture, and 4650 kilocalories of energy per kilogram. The formulations of the control and HFD/CD diets are detailed in Appendix A. A 20% fructose solution was prepared by dissolving 300 g of fructose in 1200 mL of distilled water. Both the control and HFD/CD diets were prepared at the Transgenic Animal Technologies Application and Research Center Diet Unit, utilizing purified raw materials. Each cage was provided with diets by weighing them, and the remaining diets were weighed weekly to calculate the weekly diet consumption per individual.

### 2.3. Vaccination Protocol

In the vaccination process, the hepatitis B vaccine (rDNA) [0323Y008D-Serum Institute of India Pvt. Ltd., Pune, India], commercially known as a hepatitis B vaccine containing 10 µg of HBsAg per 0.5 milliliter, was used. At the end of the 10th week, HBV vaccination was administered to both the control and NAFLD groups through three different programs. The standard-dose vaccination program (ND) was administered on days 0, 7, and 21, following the protocol reported by Yang et al. This consisted of three repeated doses, with each dose containing 4 µg of HBsAg per mouse [25]. The first high-dose vaccination program (HD2) involved administering two doses of 8 µg of HBsAg per mouse on days 0 and 7. The other high-dose vaccination program (HD3) was administered on days 0, 7, and 21, with each dose containing 8 µg of HBsAg per mouse, totaling three repeated doses. The HBV vaccination programs are shown in Appendix A.

### 2.4. Blood and Tissue Samples

One week before starting the vaccination program (at the end of the 10th week), approximately 100 μL of tail blood sample was collected from each mouse. At the end of the 16th week, all mice were euthanized via sevoflurane anesthesia by sampling blood from the axillary artery. Axillary blood samples were collected into two separate anticoagulant tubes. In the first tube, peripheral mononuclear cell isolation was performed for lymphocyte panel analysis, while in the second tube, plasma was separated by centrifugation at 5000 rpm for 15 min for the determination of anti-HB titers. Plasma and mononuclear cells were frozen at −80 °C for analysis. Peripheral mononuclear blood cells were isolated in all the blood samples collected and these cells were frozen at −80 °C and stored until lymphocyte panel analyses were conducted.

For tissue analyses, samples were taken from the liver, spleen, and thymus. Two samples were taken from each organ, with one set of samples being frozen fresh in a freezing medium (−80 °C). The other set of samples was placed in formalin for histopathological evaluation before staining with hematoxylin and eosin (H&E).

### 2.5. Analysis of Mononuclear Cells

One week before starting the vaccination programs (10th week) and at the end of in vivo experiments (16th week), blood samples were collected for the isolation of peripheral mononuclear cells. The blood samples were first pipetted with an equal volume of PBS (Phosphate Buffer Saline) and then transferred to Eppendorf tubes containing 0.5–1 mL of the lymphocyte separation medium. After centrifugation at 2500 rpm for 15 min, peripheric mononuclear cells were observed to accumulate in a halo pattern at the top of the tube. These cells were transferred to new Eppendorf tubes containing 1 cc of PBS for washing. After centrifugation at 5000 rpm for 5 min, the PBS was removed. The remaining mononuclear cell pellet was mixed with 500 μL of freezing medium (10% DMSO, 40% fetal bovine serum, and 50% RPMI medium), prepared in advance to prevent cell damage, and then frozen at −80 °C.

### 2.6. Histopatologic Evaluation

Samples from the liver, spleen, and thymus were fixed in formalin and prepared for sectioning via embedding in paraffin. Sections with a diameter of 4 µm were obtained from the paraffin blocks and stained with H&E. In order to determine the occurrence of NAFLD in the liver, samples were taken. We used the NAFLD activity score (NAS), which was designed and validated by the Pathology Committee of the NASH Clinical Research Network. The proposed NAS is the unweighted sum of steatosis, lobular inflammation, and hepatocellular ballooning scores [26,27]. Liver steatosis was scored as 0 (<5% steatosis), 1 (6–33% steatosis), 2 (34–66% steatosis), or 3 (67–100% steatosis); lobular inflammation was scored as 0 (no foci), 1 (<2 foci per 200× field), 2 (2–4 foci per 200× field), or 3 (>4 foci per 200× field); and hepatocyte ballooning was scored as 0 (none), 1 (few balloon cells), or 2 (many cells/prominent ballooning). Liver samples were also examined for fibrosis (0–4) [26,27].

### 2.7. Anti-HB Titers

The antibody levels against HBsAg (anti-HBs) in mouse plasma were measured using a AFG Bioscience Mouse Hepatitis B Virus Surface Antibody ELISA Kit via the ELISA method. Each sample was tested in duplicate. First, 50 µL of negative and positive controls were added to the kit. Next, 40 µL of the sample diluent was added to each well, except for the control wells. Then, 10 µL of the sample was added to the wells containing sample diluent. The plate was incubated at 37 °C for 30 min. The wells were washed five times with a washing solution made from 580 mL of distilled water and 20 mL of wash buffer. Following the washes, 50 µL of conjugate was added to the wells and they were incubated again at 37 °C for 30 min. After this incubation, the wells were washed five more times, and 50 µL of Chromogen A and Chromogen B solutions were added to each well. The plate was incubated at 37 °C for 15 min; then, 50 µL of the stop solution was added to halt the reaction. Finally, absorbance values were measured at 450 nm using a BIOTEK TS800 ELISA reader (Agilent, Santa Clara, CA, United State).

### 2.8. Flow-Cytometric Counting of Blood Helper T and Regulatory T Cells

One week before starting the vaccination programs (10th week) and at the end of the immunization process (16th week), peripheral blood mononuclear cells were isolated from mouse blood using the density gradient method. To each tube, 1 cc of freezing medium was added, and the samples were centrifuged at 2500 rpm for 3 min. Then, the samples were preserved at −80 °C.

Before this study, the frozen lymphocytes were brought to room temperature and added to Eppendorf tubes. The media were then removed after we incubated the tubes in an incubator for 30 min. One cc of completed culture media was added again to the tubes, and after centrifugation the media were removed. The remaining solution in the Eppendorf tubes was divided into three flow cytometry tubes for each sample, initiating the staining processes for B cell mixture (B-mix), follicular helper T cells (Tfhs), and regulatory T cells (Tregs).

Subsequently, for Tfh and Treg staining, cells were incubated in the dark at room temperature for 30 min with fluorescently labeled antibodies targeting CD4-PE/Cy7, CD25-FITC, ICOS-PerCP/Cy5.5, and CXCR5-APC. For B-mix tubes, CD19-FITC, CD38-APC, and CD27-APC markers were added. After incubation, the cells were washed once with 1X PBS and centrifuged. Following treatment with fixation and permeabilization buffers, the cells were treated in the dark at room temperature for 30 min with antibodies targeting FOXP3-PE and PD-1-APC/Cy7. After washing with 1X PBS and centrifugation, the samples were analyzed on a Beckston Dickenson FACS CANTO II flow cytometry device and the data were collected from the device for cell counting. All antibodies were obtained from Biolegend^®^, San Diego, CA, USA, and buffers were obtained from BD^®^, Franklin Lakes, NJ, USA.

### 2.9. Immunofluorescent Staining of Mouse Thymus and Spleen Sections

Fragments from the thymus and spleen of the mice were embedded in Tissue-Tek^®^ Optimum Cutting Temperature compounds immediately after removal and then cut into 20 µm sections using a cryostat set to −20 °C. The sections were transferred onto slides at room temperature and allowed to dry. After drying, the sections were fixed with 4% paraformaldehyde (PFA) at room temperature for 10 min and then washed with PBS.

For permeabilization/blocking, the sections were incubated at room temperature for 1 h in PBS containing 1% BSA and 0.5% Triton X-100 (permeabilization/blocking buffer). Subsequently, the slides were labeled with diluted primary antibodies in a PBS buffer containing 0.5% BSA and 0.25% Triton X-100 (pH 7.2). The primary antibodies used were CD19-FITC/CD38-APC, applied for the immunophenotypic characterization of plasmablast cells, and CD4^-^FITC/FOXP3-PE, used for the immunophenotyping of Tregs. After incubation, the sections were washed with PBS and then sealed with a mounting solution containing DAPI to label cell nuclei. The sealed sections were visualized under a fluorescence microscope using appropriate wavelength lasers with the JuLI Stage Real-Time Cell History Recorder.

### 2.10. Statistical Analysis

The IBM-SPSS 23.0 program was used for statistical analyses. Due to the sample size of 7 in subgroups, non-parametric tests were employed, and medians were calculated along with quartiles.

The correlation between anti-HB titers and immune parameters in the control and NAFLD groups was analyzed using Spearman’s Rho.

The variation in variables from the pre-vaccination stage to the post-vaccination stage was analyzed with the Wilcoxon Signed-Rank Test. Differences in variables between two independent groups (control and NAFLD groups) were examined using the Mann–Whitney U test. Differences in variables among three vaccination groups (ND, HD2, HD3) were examined using the Kruskal–Wallis test.

A single interaction analysis based on dietary habits and vaccination groups was performed using “Generalized Linear Models”, and estimated means were calculated along with a 95% Wald confidence interval.

A significance level of *p* < 0.05 was considered to indicate statistical significance.

## 3. Results

### 3.1. Nutritional/Growth Parameters of Experimental Groups

It was observed that the initial weights of the mice in the experimental groups were similar (Appendix A). After the initiation of the feeding protocol, it was found that the median weights of the NAFLD group were significantly higher than those of the control group from the first week onward (*p* < 0.001). Sample visuals taken from weighings conducted during the 16th week are shared in Figure 1. At both the 11th week when had vaccination started and at the end of the immunization period (16th week), there was no statistically significant difference in the body weights of the control subgroups and the NAFLD subgroups (Appendix A).

**Figure 1 vaccines-12-00934-f001:**
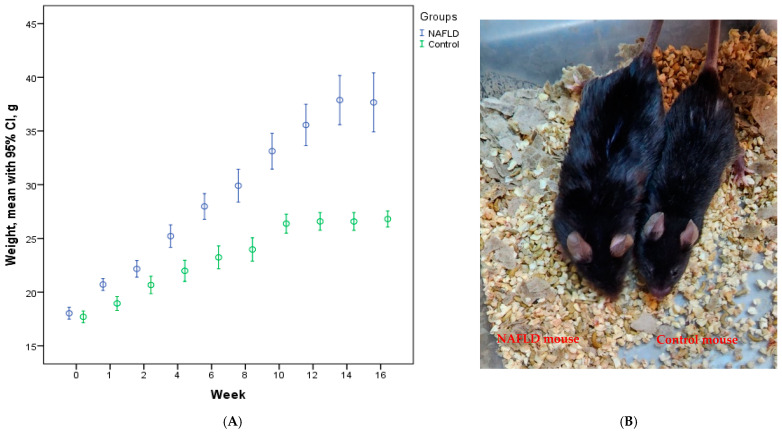
Body weight change (mean with 95% CI values) graph, structued by weeks in the experimental groups (**A**), and images representing mice from the control and NAFLD groups at the end of the 16th week (**B**).

At the end of the in vivo experiments, the levels of hepatic steatosis (Figure 2), lobular inflammation, and hepatocellular ballooning (Figure 3) in liver tissues taken from all animals were determined through histopathological examination and NAS was calculated. It was observed that 81% of the mice in the NAFLD group exhibited hepatic steatosis, while only one animal in the control group showed mild steatosis. The mean steatosis scores in the NAFLD groups were 0.86 [median (1st–3rd quartile): 1 (0–2)] for the NAFLD/ND group and 1.79 [2 (1–3)] for the NAFLD/HD2-HD3 group (*p* = 0.067). Within the NAFLD group, four mice had third-degree steatosis, six had second-degree steatosis, and seven had first-degree steatosis; in four mice, no hepatic steatosis was detected. Statistical analysis indicated a significant difference in terms of the hepatic steatosis between the control and NAFLD groups (*p* < 0.001).

**Figure 2 vaccines-12-00934-f002:**
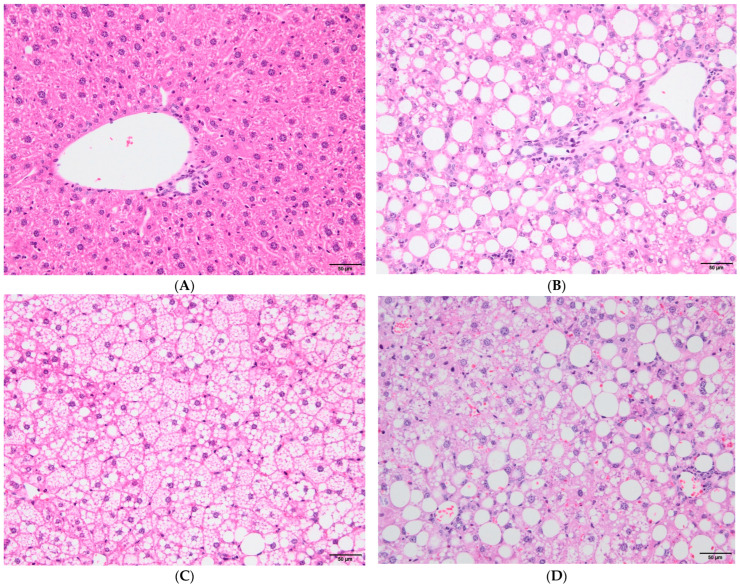
Images of liver steatosis in control and NAFLD groups (200× magnification). [(**A**) Healthy liver image, control group; (**B**) image of macrovesicular steatosis in the liver, NAFLD group; (**C**) image of microvesicular steatosis in the liver, NAFLD group; (**D**) microvesicular steatosis accompanying macrovesicular steatosis, NAFLD group].

**Figure 3 vaccines-12-00934-f003:**
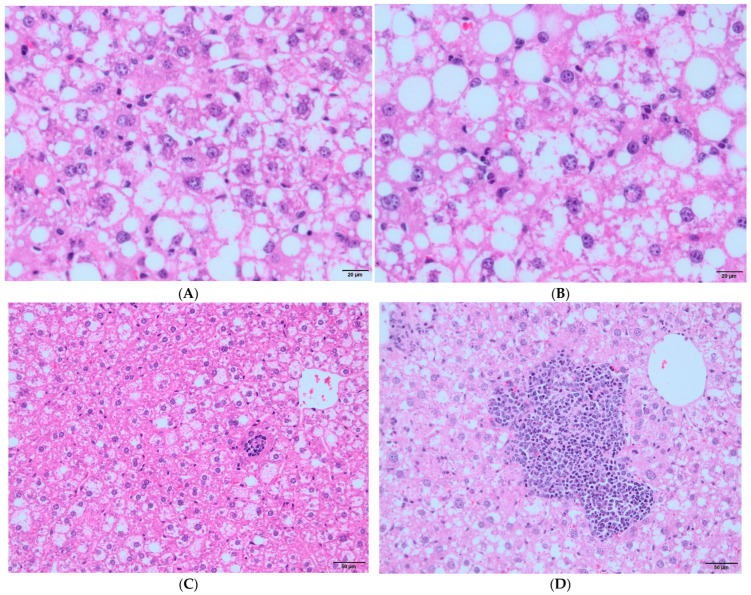
Images of morphological changes in NAFLD groups. (**A**) Hepatocytes showing ballooning degeneration (arrows, 400× magnification); (**B**) mitotic figure is observed as sign of regeneration (circle, 400× magnification); (**C**) image of lobular inflammation in = liver (200× magnification); (**D**) marked lobular inflammation around the central vein (200× magnification).

The mean [median (1st–3rd quartile)] lobular inflammation scores were 1.5 [1.5 (1–2)] in the control/ND group, 1.58 [2 (1–2)] in the control/HD2-HD3 group, 1.29 [1 (1–2)] in the NAFLD/ND group, and 1.64 [2 (1–2)] in the NAFLD/HD2-HD3 group. There was no significant difference in lobular inflammation scores between the groups (*p* = 0.809 for the control group, and *p* = 0.525 for the NAFLD group).

Hepatocyte ballooning was not detected in any mice in the control group. However, the hepatocyte ballooning scores in the NAFLD/HD2-HD3 group were significantly higher than those in the NAFLD/ND group [mean [median (1st–3rd quartile)]: 1.43 [2 (1–2)] vs. 0.71 [1 (0–1)]; *p* = 0.046].

The mean [median (1st–3rd quartile)] NAS scores were 1.50 [1.5 (1–2)] in the control/ND group and 1.67 [2 (1–2)] in the control/HD2-HD3 group (*p* = 0.616). However, the scores were 2.86 [3 (1–5)] in the NAFLD/ND group and 4.86 [5 (3.75–6.25)] in the NAFLD/HD2-HD3 subgroups (*p* = 0.056).

In the histopathological evaluation, no fibrosis was detected in any liver samples.

### 3.2. Immunological Parameters in Experimental Groups According to HBV Vaccination Program

Before vaccination, the CD27^+^ values of the NAFLD group were lower than those of the control group (*p* = 0.037), while the CD38^+^ values were higher (*p* = 0.024). No intergroup differences were observed in the levels of cells carrying both CD27^+^ and CD38^+^, nor were any found in Treg and Tfh levels (Table 1). In the NAFLD/HD3 subgroup, the level of Tregs in the blood collected at the post-immunization stage was higher than before vaccination (*p* = 0.017). No difference was observed in the control/HD3 subgroup. In the control/HD2 subgroup, the level of Tfhs in the blood collected at the post-immunization stage was higher than before vaccination (*p* = 0.010). No difference was found in the NAFLD/HD2 subgroup. The Treg/Tfh ratio did not show a statistically significant differences between the groups in terms of the blood samples taken before vaccination. There was no difference when comparing levels at the post-immunization stage with the pre-vaccination values. When the level of CD27^+^CD38^+^ in the blood samples taken at the post-immunization stage was compared with pre-vaccination values, it was low in both the control/HD2 and NAFLD/HD2 subgroups (control/HD2: *p* = 0.033, NAFLD/HD2: *p* = 0.048, Table 1).

### 3.3. Hepatitis B Vaccine Responses in Experimental Groups

Different HBV vaccination programs did not create differences in terms of anti-HB titers in both the control and NAFLD groups (Table 2). The various vaccination programs applied in our study did not result in differences in anti-HB titers between the control group and the NAFLD groups.

The two-way effects of animal groups and vaccination programs on immune parameters were examined using “Generalized Linear Models”. Different vaccination programs did not create differences in anti-HB titers between the control and NAFLD groups or among the vaccination program groups (Table 3).

The levels of CD27^+^ and CD27^+^CD38^+^ cells in the blood before vaccination were statistically significantly lower in the NAFLD group compared to the control group. However, this difference did not vary among the vaccination groups.

No statistical differences were observed in the levels of Tregs, Tfhs, and CD38^+^ cells when comparing both the pre-vaccination control and NAFLD groups and the post-immunization vaccination groups.

### 3.4. Correlation between Hepatitis B Vaccine Responses and Immunological Parameter Levels in the Control and NAFLD Groups

The correlation between hepatitis B vaccine responses and the levels of immunological parameters in the experimental groups were evaluated. When all three vaccination protocols were considered together, no correlation was found between anti-HB titers and immune parameters in the control groups. This suggests that, in the control groups, there was no clear relationship between antibody levels and the measured immune parameters. In contrast to the control groups, a negative relationship was observed between Tfh levels and CD27^+^ cell levels and antibody levels in the NAFLD groups when comparing values before the vaccination period and after the immunization period. This implied that higher levels of Tfh and CD27^+^ cells were associated with lower antibody levels. On the other hand, there was a positive relationship between Tregs/Tfhs and CD38^+^ cell levels and antibody levels. This suggested that higher levels of Tregs/Tfhs and CD38^+^ cells were associated with higher antibody levels. There was no correlation found between Tregs and CD27^+^CD38^+^ cell levels and antibody levels. This indicated that these specific combinations of immune cell levels did not have a clear relationship with antibody levels (Table 4).

**Table 4 vaccines-12-00934-t004:** Correlation between anti-HB titers and immune parameter levels, Spearman’s Rho.

	Control Groups		NAFLD Groups	
	r_s_	*p*	r_s_	*p*
Treg	−0.18	0.499	−0.02	0.931
Tfh	−0.21	0.422	−0.47	0.030
Treg/Tfh	−0.21	0.444	0.67	0.002
CD27^+^	−0.32	0.202	−0.59	0.005
CD38^+^	0.07	0.769	0.64	0.002
CD27^+^CD38^+^	0.10	0.698	−0.04	0.849

Treg: regulatory T cell; Tfh: follicular helper T cell.

### 3.5. Immunophenotyping in Experimental Groups

When examining the HBV vaccine responses in all vaccination program groups, a seropositivity of 89.7% was found. To support this finding, immunophenotyping with immunofluorescent staining was performed on the spleen and thymus tissues of randomly selected seropositive mice from the groups. In randomly selected mice with codes of KND2, FND2, and KHD2-3, specific staining for both CD19^+^ (green) and CD38^+^ (red) cells was observed in the thymus, with dense lymphocyte islands, widespread mitotic divisions, and proliferation specifically being detected in the cortex region (X20 objective) (Figure 4). Immunophenotyping images of CD19^+^ and CD38^+^ cells in the spleen tissue are presented in Figure 5 (X20 objective).

**Figure 4 vaccines-12-00934-f004:**
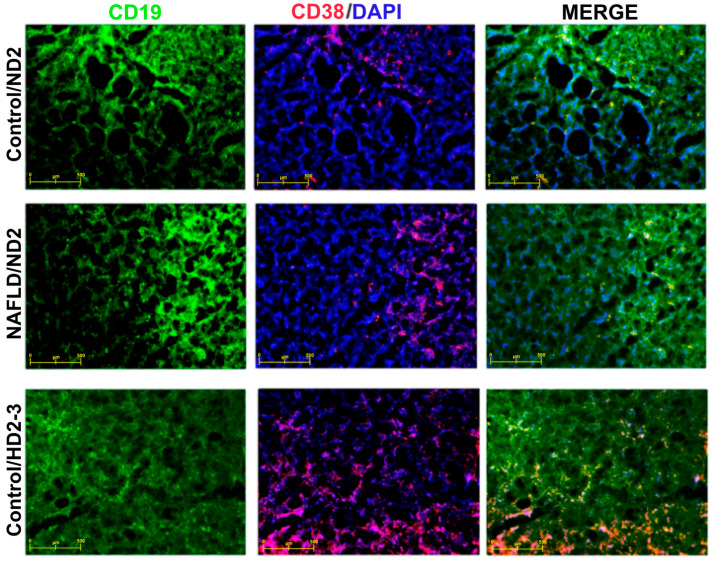
Immune phenotyping of CD19^+^ and CD38^+^ in thymus tissue of control/ND-2, NAFLD/ND-2, and control/HD2-3 mice.

## 4. Discussion

In our study, blood samples taken before vaccination programs in mice using an established NAFLD model showed lower CD27^+^ cell counts and higher CD38^+^ cell counts compared to healthy mice. However, post-immunization anti-HB titers were similar across all vaccination subgroups in both the control and NAFLD groups. Nonetheless, post-immunization blood samples revealed higher Treg levels in the NAFLD/HD3 subgroup and higher Tfh levels in the control/HD2 subgroup compared to pre-vaccination levels. In the NAFLD/ND subgroup, post-immunization CD27^+^ cell levels were lower, while CD38^+^ cell levels were higher compared to the pre-vaccination levels.

In obese individuals, due to increased body fat and leptin production, an effective immune response against vaccines or infections may not develop. The relationship between obesity and a weak immune response to vaccines was first observed in 1985, when hepatitis B vaccines were administered to obese hospital workers [14]. An increasing number of scientific publications suggest that obese individuals may exhibit lower antibody responses to vaccines compared to non-obese individuals [12,13,28,29,30,31]. Additionally, studies have shown that obese mice experience a faster decline in neutralizing antibody titers compared to normal mice, which may impair the memory T cell response in obese individuals [32]. This suggests that there will be a diminished efficacy of adaptive B and T cell in cases of obesity [33,34,35]. In obese mice, there is a noted decrease in key transcripts involved in the early lymphoid response, likely due to defects in the bone marrow environment [36]. Obesity is also linked to the increased activation of pro-inflammatory Th1 and Th17 cells and a reduction in anti-inflammatory Th2 cells and Tregs [37].

There is limited research on HBV vaccine responses in NAFLD cases [18]. Joshi et al. [18] reported that the HBV vaccine immune response was lower in mice with NAFLD compared to healthy mice. Miyake et al. [38] found that dendritic cells from mice fed on a high-fat diet exhibited impaired functions, resulting in a reduced immune response to HBsAg and HBcAg.

In our study, a statistically non-significant decrease in the number of Tregs was observed in the NAFLD group compared to the control group in the blood samples taken before vaccination. Previous research has reported a significant decrease in splenic Tregs in mice using established NAFLD models [39]. Various studies on animal obesity have also noted a reduction in Tregs in both the spleen and visceral adipose tissue [40,41]. Additionally, our study found high levels of plasma cells carrying CD38 in the pre-vaccination blood of NAFLD mice, which is notable. CD38 is a receptor expressed in regulatory B (Breg) cells. Measuring the expression levels of CD38^+^ cells in peripheral blood of HBV-infected patients helps to objectively reflect their cellular immune function status and serves as a reference for assessing disease progression and treatment efficacy [42].

In our study, a significant increase in the percentage of peripheral Tregs was observed in the NAFLD/HD3 subgroup post-immunization compared to the pre-HBV vaccination period. This situation may suggest that the three high doses of HBV vaccination administered in the NAFLD/HD3 subgroup triggered inflammation in these mice, who already had fatty livers. Tregs, a subset of CD4^+^ T cells, play a role in inhibiting excessive responses to the virus and suppressing excessive inflammation. The increase in Treg levels in NAFLD mice may be related to an inflammatory response. It has been reported that chronic EBV infection increases the proliferation of Tregs and suppresses the immune response to HBV [43,44,45]. In the study conducted by Wolf et al. [46], the authors reported that there were no changes in Treg frequencies and phenotypes after influenza vaccination. However, after hepatitis B vaccination, both resting-Treg and activated-Treg subpopulations showed a slight increase in frequencies and a decrease in the expression of CD39, a functional marker of activated Tregs. In the same study, they also reported a decrease in the frequency of resting Tregs and an increase in the expression of the activation marker CD25 in both subpopulations after live attenuated vaccination. This was explained by the potential transformation of resting Tregs into activated Tregs due to vaccine virus replication [46]. CD4^+^ T cells exhibit plasticity in chronic inflammation and situations with excessive inflammation, transforming into regulatory T cells (Treg) and reducing the number of inflammatory cells [47].

Another important subset of CD4^+^ T cells, crucial in the response to HBV infection, is that of Tfhs. Tfhs are responsible for balancing humoral immune responses carried out through antibody-producing B cells. Disturbances in antiviral B cell responses have been identified in chronic HBV infections. Additionally, these cells are reported to play an active role in B cells’ antibody production against HBV [48]. In our study, Tfh cell levels in the post-immunization period were significantly increased in the control/HD2 subgroup compared to the pre-vaccination period.

Recent studies have shown that regulatory B cells (Bregs) help to maintain immunological homeostasis [49,50,51]. Bregs induce the stimulation of Tregs by disrupting T cell differentiation. Moreover, they prevent immunopathology, particularly by suppressing pro-inflammatory cells through regulatory cytokines such as IL-10, TGF-ß, and IL-35 [49,50,52]. CD24+/highCD27+ and CD24highCD38high Bregs can suppress effector CD4^+^ T cells and dendritic cells, while Bregs carrying CD24highCD38high can induce Tregs and suppress virus-specific CD8^+^ T cells [53,54]. In our study, it was observed that both CD27^+^ and CD27^+^CD38^+^ cell levels were lower in the NAFLD group compared to the control group in the blood samples taken before vaccination. The CD27^+^ cell level showed a negative correlation with vaccine antibody titers, and the CD38^+^ cell level showed a positive correlation. There are controversial reports for Bregs and vaccine response. Bolther et al. [55] observed a decrease in IL-10 producing Bregs in high responders post-vaccination, but found no correlation between Bregs levels, IFN-g responses, and anti-HB titers. They reported that Bregs or IFN-g-positive T cells did not influence HBV vaccine responses and Breg levels were not predictive of serological responses to the HBV vaccine. However, Körber et al. [51] found elevated frequencies of CD24highCD38high Bregs in non-responders compared to responders and noted lower IL-10 expression levels in non-responders. They also reported that, compared to cases with a positive response to the HBV vaccine, cases with an insufficient response had significantly higher frequencies of CD24highCD38high Bregs on day 0 (*p* = 0.004) and day 28 (*p* = 0.012) of the second vaccination, but not on day 7 (*p* = 0.051) [51]. This variation may be explained by the possibility that the level of Bregs could change depending on the day of the serological study, such as in our study. In our study, following the first vaccine treatment in mice with NAFLD model, Breg percentages were slightly increased. However, in the second treatment with the vaccine, the Breg ratio was shown to be reduced. In addition, Bolther et al. [55] did not address the effect of booster vaccination directly. Körber et al. [51] reported a significant decrease in Breg frequencies following booster vaccination and stable IL-10 levels, implying potential improvements in vaccine efficacy in non-responders.

The evaluation of peripheral blood regulatory and inflammatory cells using flow cytometry, along with the presence of CD38^+^ B cells in the spleen and FOXP3^+^ Tregs in the thymus through immunofluorescent staining, supports the immune responses obtained.

In our study, lobular inflammation scores were similar in both the control and NAFLD groups of mice administered high-dose HBV vaccines. However, in mice fed on the same NAFLD diet, those given high doses of the HBV vaccine (HD3 and HD2) showed significantly higher hepatocyte ballooning scores compared to the ND vaccine group. The NAS scores of NAFLD mice receiving high-dose vaccines were also slightly higher than those in the normal-dose group, although this was not statistically significant (*p* = 0.056). The vaccine preparation contained 0.25–0.40 mg/0.5 mL of aluminum hydroxide, resulting in an intraperitoneal dose of 0.20–0.32 mg (6–8 mg/kg-mouse) for high-dose vaccination. The low dose was 3–4 mg/kg-mouse. Aluminum exposure is known to potentially disrupt lipid metabolism, increase fat accumulation, induce lipid peroxidation, cause oxidative stress, and trigger apoptosis in hepatocytes [56,57,58]. Moreover, the presence of steatosis may impair the liver’s detoxification capacity, potentially leading to the accumulation of metals and other toxins. This can contribute to the progression of NAFLD. This exacerbation of NAFLD can potentially accelerate the progression from simple steatosis to more severe forms such as NASH, which involves inflammation and liver cell damage. Similar to our findings, de Souza et al. [56] reported that oral gavage administration of aluminum hydroxide (10 mg/kg) three times a week for six months in BALB/c mice resulted in hepatocyte vacuolization and higher steatosis scores, but no increase in inflammatory foci in the liver parenchyma. Additionally, Hamza et al. demonstrated an increase in the percentage of apoptotic cells with a rising aluminum concentration, though smaller than the combined effect of hepatitis B antigen and aluminum in the whole vaccine, along with hepatocyte vacuolization [57].

The observation of hepatocyte ballooning in NAFLD mice, but not in mice with healthy livers treated at the same high doses, suggests that the amount of adjuvant used in high-dose vaccinations should be carefully considered. The lack of increased vaccine efficacy with higher doses also suggests that using normal doses may be safer in NAFLD cases. Vaccine adjuvants, particularly chemical additives like aluminum hydroxide, are used to enhance the immune response to vaccine antigens but may have potentially toxic mechanisms [59]. Aluminum adjuvants, when used at current doses, are generally considered safe. However, in patients with fatty liver disease, caution is necessary regarding aluminum’s toxic effects.

Our study demonstrated differences in immune responses between mice with induced NAFLD and those with a healthy liver, while showing the generation of antibodies against the HBV vaccine in all vaccination groups. This suggests that an adverse effect on clinical HBV vaccine response is not expected in individuals with NAFLD, and the same vaccination program can be applied to these cases as that used for individuals with a healthy liver. However, further studies are needed to compare HBV vaccine responses in cases of non-alcoholic steatohepatitis (NASH), where advanced liver steatosis is accompanied by inflammation and fibrosis, with the results of our study.

### Strengths and Difficulties

Due to the national HBV vaccination program in our country, it was not feasible to obtain homogeneous NAFLD cases without HBV vaccination from human subjects. Therefore, we conducted the research using animal models. By employing the C57BL-6 inbred mouse line, we minimized individual genetic variation, which is challenging to control in human subjects. Animal experiments allowed us to create the specific, uniform conditions essential for our study, which are difficult to achieve in humans.

With the in vivo experiments, the histopathological examination at the 16th week revealed that NAFLD did not manifest in 19% (*n* = 4) of the mice consuming an HFD/CD diet with 20% fructose, while varying degrees of fatty liver were observed in the remaining mice. Our study noted that the body weight of mice consuming the HFD/CD diet with 20% fructose continued to increase until the 14th week. By initiating the vaccination regimen at the 14th week instead of the 10th week and ending the in vivo experimental phase at the 19th week, the NAFLD group’s mice were allowed to consume the HFD/CD diet with 20% fructose for an additional four weeks. This extension will enable a more comprehensive assessment of NAFLD in mice. In future investigations, commencing the vaccination protocol during the week when the animals reach peak weight (as observed in our study at the 14th week) may better elucidate potential differences in vaccine efficacy. In addition, the utilization of an in vivo imaging system capable of accurately assessing the extent of fatty liver in mice would greatly enhance NAFLD studies. The absence of an unvaccinated NAFLD group in our study was a limitation. We had no information about the condition of the liver before vaccination. However, while no histopathological differences were observed in the livers of healthy mice compared to different HBV vaccine doses, the differences observed in NAFLD-induced and high-dose vaccine groups compared to the normal-dose group suggest that the changes occur when NAFLD and high-dose vaccination occur together. When the NAS values were compared among groups with 6–7 mice, borderline significance was observed. Increasing the sample size in future studies may reveal a significant difference in this parameter.

Our study’s strength lies in examining the effectiveness of three different vaccine protocols to investigate the effect of varying doses and frequencies on vaccine efficacy. Due to the absence of a standardized HBV vaccine dosage for mice, the dosage was determined by reviewing previous publications employing the HBV vaccine. However, the literature on HBV vaccine usage in mice is limited. Future studies could more effectively reveal vaccine efficacy differences by incorporating lower-dosage vaccination groups.

## 5. Conclusions

In conclusion, our study successfully established a NAFLD model in mice following a 10-week feeding regimen with a choline-deficient 45% high-fat diet (HFD) along with 20% fructose water, as evidenced by liver histopathology. Notably, the NAFLD groups exhibited more than a twofold increase in body weight compared to the control groups by the 16th week. Furthermore, our investigation revealed similarities in the HBV vaccine response between the control and NAFLD groups across three distinct vaccination protocols. Importantly, there was no difference in the HBV vaccine response among the groups regarding the HBV vaccine programs employed. Regarding immune parameters, post-immunization Tregs were notably higher in the NAFLD groups, particularly in the high-dose vaccine group (HD3). Additionally, both NAFLD and control groups exhibited a decrease in CD27^+^38^+^ cells post-vaccination compared to pre-vaccination, particularly in the two high-dose vaccine groups (HD2). The observation of high hepatocyte ballooning scores in NAFLD mice following high-dose vaccination suggests that even safe levels of vaccine adjuvants may have toxic effects in cases of steatosis. This finding leads to the conclusion that double-dose vaccination may not be effective or safe in NAFLD cases. In summary, our findings indicate that HBV vaccination elicits an antibody response in individuals with NAFLD, while also highlighting differences in immune parameter levels between NAFLD and control groups. These insights contribute to our understanding of the interplay between HBV vaccination and NAFLD, emphasizing the importance of considering immune responses in NAFLD-afflicted populations in vaccination strategies.

## Figures and Tables

**Figure 5 vaccines-12-00934-f005:**
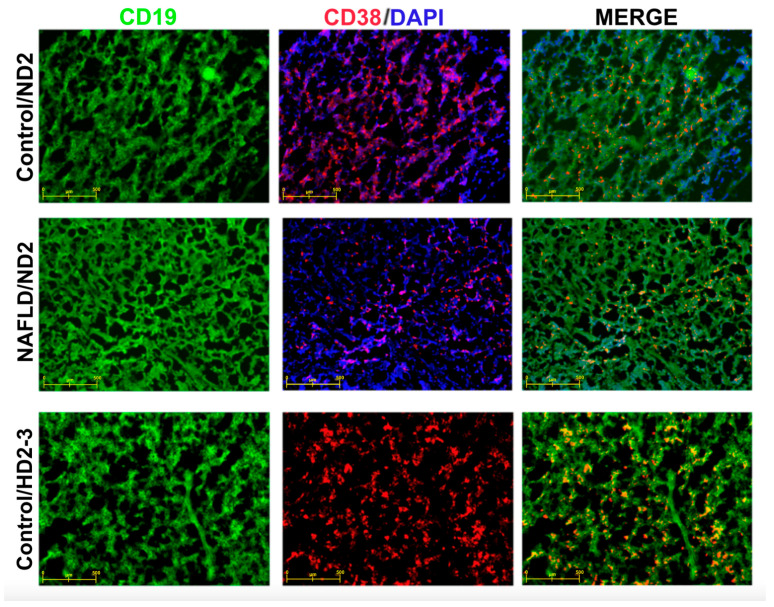
Immune phenotyping of CD19^+^ and CD38^+^ in spleen tissue of control/ND2, NAFLD/ND2, and control/HD2-3 mice.

**Table 1 vaccines-12-00934-t001:** The effects of different vaccination programs on the levels of immune parameters.

			Pre-Vaccination, PV (10th Week)		ND		HD2		HD3	Sign, *p*			
	Diet Types	*n*	Median (25th–75th Percentile)	*n*	Median (25th–75th Percentile)	*n*	Median (25th–75th Percentile)	*n*	Median (25th–75th Percentile)	PV vs. ND ^&^	PV vs. HD2 ^&^	PV vs. HD3 ^&^	ND-HD2-HD3 ^$^
Treg	Control	18	10.07 (7.41–11.95)	5	11.70 (7.30–19.30)	6	9.58 (5.13–16.20)	5	12.30 (8.38–18.90)	0.446	0.923	0.150	0.673
	NAFLD	21	7.04 (4.84–11.95)	7	9.56 (7.76–12.20)	7	7.31 (6.78–13.90)	7	13.90 (10.50–16.90)	0.155	0.321	0.012	0.138
	*p* ^#^		0.321		0.755		1.000		1.000				
Tfh	Control	17	2.81 (1.13–3.54)	5	3.48 (2.59–4.94)	6	5.67 (4.28–6.46)	6	3.55 (2.16–5.22)	0.319	0.010	0.201	0.146
	NAFLD	21	3.84 (2.81–4.28)	7	2.74 (1.47–4.60)	7	3.55 (2.01–9.68)	7	2.48 (0.00–6.06)	0.376	0.979	0.499	0.598
	*p* ^#^		0.064		0.432		0.366		0.628				
Treg/Tfh	Control	17	3.87 (1.36–12.17)	5	3.21 (1.93–6.48)	6	1.81 (1.38–2.29)	6	3.31 (2.13–6.53)	0.880	0.135	0.940	0.217
	NAFLD	21	1.95 (1.11–5.16)	7	3.97 (1.53–3.55)	7	3.33 (1.53–3.55)	7	3.21 (1.98–3.92)	0.189	0.756	0.642	0.426
	*p* ^#^		0.199		0.755		0.295		0.730				
CD27^+^	Control	18	14.55 (6.98–60.40)	6	9.99 (5.04–20.20)	6	11.48 (3.19–20.88)	6	14.15 (2.90–27.33)	0.343	0.343	0.454	0.999
	NAFLD	21	6.50 (1.95–18.75)	7	4.17 (2.16–7.14)	7	6.68 (3.23–9.09)	7	7.14 (0.75–15.00)	0.499	0.466	0.533	0.816
	*p* ^#^		0.037		0.073		0.366		0.295				
CD38^+^	Control	18	47.25 (9.69–78.30)	6	69.20 (48.00–78.40)	6	70.80 (56.28–79.90)	6	55.95 (35.03–69.18)	0.537	0.251	0.923	0.258
	NAFLD	21	78.10 (67.70–83.75)	7	85.40 (65.20–92.10)	7	73.90 (64.50–84.70)	7	72.30 (20.00–84.40)	0.155	0.876	0.717	0.305
	*p* ^#^		0.024		0.138		0.534		0.234				
CD27^+^CD38^+^	Control	18	15.15 (9.95–18.98)	6	11.10 (8.96–14.13)	6	9.20 (6.58–11.55)	6	19.90 (7.07–26.03)	0.177	0.033	0.673	0.244
	NAFLD	21	11.10 (10.15–13.90)	7	8.96 (3.78–14.20)	7	5.65 (2.78–12.90)	7	9.77 (0.00–13.10)	0.155	0.048	0.249	0.791
	*p* ^#^		0.112		0.366		0.534		0.181				

^&^ Wilcoxon Signed-Rank Test; ^$^ Independent samples Kruskal–Wallis test; ^#^ Mann–Whitney test; Treg: regulatory T cell; Tfh: follicular helper T cell.

**Table 2 vaccines-12-00934-t002:** The effects of different vaccination programs on anti-HB titers (IU/mL) in control and NAFLD groups *.

Vaccination Program	*n*	Control Groups	*n*	NAFLD Groups	*p* ^#^
ND	6	48.1 (18.2–62.3)	7	26.5 (24.5–66.9)	0.503
HD2	6	22.5 (7.8–49.4)	7	39.7 (21.2–44.7)	0.833
HD3	6	26.5 (17.8–40.4)	7	26.3 (19.5–74.5)	0.295
*p* ^$^		0.421		0.911	

* Median (25th–75th percentile); ^$^ independent samples Kruskal–Wallis test; ^#^ Mann–Whitney test.

**Table 3 vaccines-12-00934-t003:** Interaction analysis of the effects of different vaccination programs on hepatitis B vaccine response and immune parameter levels in control and NAFLD groups using Generalized Linear Models.

	Model Groups (A)		Vaccination Protocol (B)			Sign, *p*		
	Control Group	NAFLD Group	ND	HD2	HD3	A	B	AXB
*n*	18	21	13	13	13			
Anti-HB titers, IU/mL	37.0 [28.7–45.4]	32.5 [23.5–41.4]	41.4 [30.8–52.0]	29.2 [18.6–39.8]	33.6 [23.0–44.2]	0.462	0.274	0.555
Treg	12.3 [9.9–14.7]	11.8 [9.7–13.9]	12.3 [9.5–15.1]	10.4 [7.7–13.1]	13.6 [10.8–16.4]	0.756	0.267	0.903
Tfh	4.37 [3.29–5.44]	3.60 [0.63–4.56]	3.34 [2.05–4.64]	5.14 [3.91–6.80]	3.46 [2.23–4.69]	0.297	0.078	0.983
Treg/ Tfh	3.34 [2.07–4.62]	3.77 [2.53–5.01]	4.69 [3.21–6.18]	2.39 [0.98–3.81]	3.58 [1.88–5.29]	0.637	0.089	0.528
CD27^+^	13.6 [10.0–17.2]	6.2 [2.9–9.6]	9.2 [5.0–13.5]	9.2 [5.0–13.5]	11.3 [7.1–15.6]	0.003	0.735	0.967
CD38^+^	59.9 [49.6–70.2]	70.5 [61.0–80.0]	70.6 [58.5–82.7]	69.2 [57.1–81.4]	55.8 [43.7–67.9]	0.138	0.173	0.528
CD27^+^ CD38^+^	12.3 [9.6–15.1]	8.1 [5.6–10.7]	10.1 [6.9–13.3]	7.9 [4.6–11.1]	12.7 [9.5–15.9]	0.027	0.113	0.280

Estimated mean [95% Wald confidence interval]; Treg: regulatory T cell; Tfh: follicular helper T cell.

## Data Availability

The datasets used and/or analyzed during the current study are available from the corresponding author on reasonable request.

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
