# Peer review of "Investigation of the Hepatitis-B Vaccine’s Immune Response in a Non-Alcoholic Fatty Liver Disease Mouse Model"

_vaccines, 2024, doi:10.3390/vaccines12080934_

Round 1
Reviewer 1 Report
Comments and Suggestions for Authors
This manuscript is a commendable piece of work, demonstrating a strong understanding of the subject matter. While there are a few grammatical errors that need attention, such as Line 24 where 'Than' should be replaced with 'Then ', the authors have done a good job overall.
He documents some errors in the figures, such as in Figure 1, where the dotted lines do not appear straight. Figure 1 must be made with panel A and panel B. In addition, the y-axis legend needs to be read, and the text is shown in a pile. Authors should be more careful when presenting their results graphically. Although it may be obvious, the image of the animals must be explained correctly, such as indicating which treatment each animal corresponds to. In Figure 2, the scale bar does not resemble the size and is not stated in the figure caption—the same in Figures 3 and 4. The figure captions should be revised because typographical errors are observed.
The authors present a well-supported introduction, and the discussion contains relevant information in the context of the work.
Author Response
Comments1: This manuscript is a commendable piece of work, demonstrating a strong understanding of the subject matter.
Responce 1: Thank you for your kind words. We are grateful for your positive feedback and are pleased that our manuscript demonstrates a strong understanding of the subject matter.
Comments 2: While there are a few grammatical errors that need attention, such as Line 24 where 'Than' should be replaced with 'Then ', the authors have done a good job overall.
Responce 2: Agree. The relevant sentence has been completely revised in line 24 with other reviewer comments as “All mice were euthanized, blood samples for anti-HBs titers, T follicular helper, T follicular regu-latory, CD27+, CD38+ cells, and liver, spleen and thymus for histopatologic evaluation were taken.”
Comments 3: He documents some errors in the figures, such as in Figure 1, where the dotted lines do not appear straight. Figure 1 must be made with panel A and panel B. In addition, the y-axis legend needs to be read, and the text is shown in a pile. Authors should be more careful when presenting their results graphically. Although it may be obvious, the image of the animals must be explained correctly, such as indicating which treatment each animal corresponds to.
Responce 3: Agree. Figure 1 was designed with parts A and B. The dotted lines were removed. Part A of Figure 1 has been revised as recomended and moved to Supplementary file. A new Figure 1 depicting body weight changes over weeks was designed with mean and 95% CI, as recommended by Reviewer 2
We have revised Figure 1, part B, the animals were coded as Control and NAFLD mouse
Thank you for your constructive criticism. We appreciate your feedback and will be more careful in presenting our results graphically to enhance the quality of our work.
Comments 4: In Figure 2, the scale bar does not resemble the size and is not stated in the figure caption—the same in Figures 3 and 4. The figure captions should be revised because typographical errors are observed.
Responce 4: Agree. We have revised Figure 2, new images were captured, and scale bars were added. Additionally, scale bars have been included in Figures 3 and 4 (a new Figure was added and named as Figure 4 and 5).
Comments 5: The authors present a well-supported introduction, and the discussion contains relevant information in the context of the work.
Responce 5: Thank you for your positive feedback. We appreciate your recognition of our well-supported introduction and relevant discussion.

Reviewer 2 Report
Comments and Suggestions for Authors
This study by TuÄŸba KÜTÜK et al. provided additional knowledge about whether Non-Alcoholic Fatty Liver Disease (NAFLD) affects the HBV vaccine efficacy in a mouse model. The overall science is good and the text flows. Some errors and language mistakes exist. Below are some examples
Line 189: “For Bmix tubes, 0.5 µl of C19,” CD19.
Line 181 and 199: Inconsistence between “FACS CANTO II”and “Facs Canto II”
Line 189 to 197: The origin of the FACS related antibody and corresponding fluorophores were not indicated. Only mentioned in Line 211 -212, but incomplete.
PAGE 6: Figure 1 left panel. I would suggest the author to represent the error bar for each group, which would be obvious seen the body weight differences between Control and NAFLD groups.
PAGE 6: for Figure 2, there is no indication of the scale bar but presented within the figure. Compared to magnification, the scale bar is preferred in academic writing. Which Figure 3 and 4 on Page 10 share similar issue.
Line 400-405: When citing the reference Bolther et al. [56], its conclusion seems to contradict the reports by Körber et al. [52], adding details for the Bolther et al. related research may provide more information for readers to compare differences between two studies.
Comments on the Quality of English LanguageFew errors and language mistakes exist
Author Response
Comments 1: This study by TuÄŸba KÜTÜK et al. provided additional knowledge about whether Non-Alcoholic Fatty Liver Disease (NAFLD) affects the HBV vaccine efficacy in a mouse model. The overall science is good and the text flows.
Response 1: Thank you for your feedback. We are pleased that you found our study on the impact of Non-Alcoholic Fatty Liver Disease (NAFLD) on HBV vaccine efficacy in a mouse model to be informative. We appreciate your positive comments on the overall science and the flow of the text.
Comment 2: Some errors and language mistakes exist. Below are some examples Line 189: “For Bmix tubes, 0.5 µl of C19,” CD19.
Response 2: Agree. 'C19' corrected as 'CD19' (Page 5, Line 200).
Comment 3: Line 181 and 199: Inconsistence between “FACS CANTO II”and “Facs Canto II”
Response 3: Agree. 'Facs Canto II' corrected as 'FACS CANTO II' (Page 5, line 205).
Comment 4: Line 189 to 197; The origin of the FACS related antibody and corresponding fluorophores were not indicated. Only mentioned in Line 211 -212, but incomplete.
Response 4: Agree. We have revised this part as “Subsequently, for Tfh and Treg cell staining, cells were incubated in the dark at room temperature for 30 minutes with fluorescently labeled antibodies targeting CD4-PE/Cy7, CD25-FITC, ICOS-PerCP/Cy5.5, and CXCR5-APC. For B-mix tubes, CD19-FITC, CD38-APC, and CD27-APC markers were added. After incubation, the cells were washed once with 1X PBS and centrifuged. Following treatment with fixation and permeabilization buffers, the cells were treated in the dark at room temperature for 30 minutes with antibodies targeting FOXP3-PE and PD-1-APC/Cy7. After washing with 1X PBS and centrifugation, the samples were analyzed on a Beckston Dickenson FACS CANTO II flow cytometry device and the data were collected from the device for cell counting. All antibodies were obtained from Bioegend®, USA and buffers were obtained from BD®, USA.” (Page 5, Paragraph 1).
Comment 5: PAGE 6: Figure 1 left panel. I would suggest the author to represent the error bar for each group, which would be obvious seen the body weight differences between Control and NAFLD groups.
Response 5: Agree. Part A of Figure 1 has been moved to Supplementary file. A new Figure 1 (page 6) depicting body weight changes over weeks was designed with mean and 95% CI according to experimental groups, as recommended.
Comment 6: PAGE 6; for Figure 2, there is no indication of the scale bar but presented within the figure. Compared to magnification, the scale bar is preferred in academic writing. Which Figure 3 and 4 on Page 10 share similar issue.
Response 6: Agree. We have captured new images, and added scale bars for Figure 2. Additionally, scale bars have been included in Figures 3 and 4 (a new Figure was added and named as Figure 4 and 5).
Comment 7: Line 400-405: When citing the reference Bolther et al. [56], its conclusion seems to contradict the reports by Körber et al. [52], adding details for the Bolther et al. related research may provide more information for readers to compare differences between two studies.
Response 7: Agree. This part is revised as “There are controversial reports for Bregs and vaccine response. Bolther et al. [55] observed a decrease in IL-10 producing Bregs in high responders post-vaccination but found no correlation between Bregs levels, IFN-g responses, and anti-HBs titers. They reported that Bregs or IFN-g positive T cells did not influence HBV vaccine response and Breg cell levels were not predictive of serological responses to the HBV vaccine. However, Körber et al. [51] ] found elevated frequencies of CD24highCD38high Bregs in non-responders compared to responders and noted lower IL-10 expression levels in non-responders. They also reported that compared to cases with a positive response to the HBV vaccine, cases with an insufficient response had significantly higher frequencies of CD24highCD38high Breg cells on day 0 (p = 0.004) and day 28 (p = 0.012) of the second vaccination, but not on day 7 (p = 0.051) [51]. This variation may be explained by the possibility that the level of Bregs could change depending on the day of the serology study, such as in our study. In our study, following first vaccine treatment in mice with NAFLD model, Breg cell percentages were slightly increased however in second treatment with vaccine, Breg cell ratio were shown to be reduced. In addition, Bolther et al. [55] did not address the effect of booster vaccination directly. Körber et al. [51] reported a significant decrease in Bregs frequencies following booster vaccination and stable IL-10 levels, implying potential improvement in vaccine efficacy in non-responders.” (Page 15, Line 440-457).
Comment 8: Comments on the Quality of English Language
Response 8: Agree. Proofreading has been performed.

Reviewer 3 Report
Comments and Suggestions for Authors
The authors have done experiments in mouse with normal livers and a mice where they created non-alcoholic fatty liver. Vaccination with an hepatitis B vaccine induced anti-Hbs in all, but in the NAFLD group differences in CD27+ and CD27+CD38 profiles were identified (thymus, spleen were examined) that may pleay a role in immunogenicity of vaccins.
Comments
1.It is for this reviewer somewhat difficult to understand the conceptual design of this study (also unclear if any clinical input was obtained, or all laboratory scientists?)
In reading the article, it is hard to escape the impression that the primary focus was anti-body titers and that subsequently without a clear pathobiological design certain aspects of the immune responses were compared in relatively small groups of mice taking in account subcategories. More phishing than design unless more clearly explained to the reader. Some attempt is done in the discussion but I cannot easily detect a clear logic from aim of the study towards conclusions. Help the reader, certainly the less educated immunologist to better understand.
It is interesting that the ethics committee approved the protocol although from the introduction it remains initially unclear why animals were needed from data that more meaningful could have been obtained from humans. The authors did various more detailed investigations that look - as outlined - more like an add on than a clear conceptual design as outlined and therefore the animals needed to be submitted to experiments.
It is my impression that the authors use often immunity where I suggest they use immune response and in some situations immunogenicity would appear to be a more appropriate term
Vaccination for HBV is for 2 reasons important in patients with chronic liver disease (see for example review Joshi et al, 2021 etc.) 1. Excessive mortality and morbidity in case of (any) underlying liver disease. (Therefore also HAV vaccination important). 2. Prevention of HBV and the spectrum of related pathology including risk HCC
It remains somewhat puzzling why the authors study in detail certain aspects of the immune response but do not describe any further detail of pathology. It is hard to see (image quality?) if, for example, any inflammatory response or fiborisis in the NAFLD livers has occurred. Only H and E stains?(difficult to see in (b) specimen: steatohepatitis? The response various tremendously among humans. Therefore the "pure storage of fat" (steatosis) is different from significant inflammation (steatohepatitis)
Abstract
do we need to know specifics of diet in abstract?
Introduction
can be shortened and focus more on presumed pathobiology
patients with advanced liver disease tend to be poorer vaccine responders in general than healthy people also patients with chronic diseases
the authors rightly note the unpredictability of responses due to many variables: are their groups big enough for detailed analysis?
The high risk groups are?
M and M:
is all detail necessary (like determination of antibody titers etc etc.)?
Minor
3500 kcal etc is the total calories fed to the animals during the study period if correctly understood
Author Response
Comments 1: The authors have done experiments in mouse with normal livers and a mice where they created non-alcoholic fatty liver. Vaccination with an hepatitis B vaccine induced anti-Hbs in all, but in the NAFLD group differences in CD27+ and CD27+CD38 profiles were identified (thymus, spleen were examined) that may pleay a role in immunogenicity of vaccins.
Response 1: Thank you for your detailed feedback. We appreciate your recognition of our experiments comparing mice with normal livers to those with non-alcoholic fatty liver. We are glad to hear that you found our findings on the differences in CD27+ and CD27+CD38+ profiles, which may play a role in the immunogenicity of the HBV vaccine, to be significant.
Comments 2: It is for this reviewer somewhat difficult to understand the conceptual design of this study (also unclear if any clinical input was obtained, or all laboratory scientists?)
Response 2: The design of laboratory animal studies was used. This includes details on the characteristics of the animals used, the diet administered, the vaccination protocol, the samples collected, and the laboratory analyses performed.
Comments 3: In reading the article, it is hard to escape the impression that the primary focus was anti-body titers and that subsequently without a clear pathobiological design certain aspects of the immune responses were compared in relatively small groups of mice taking in account subcategories.
Response 3: In order to enhance clarity and comprehensiveness, the aim statement has been revised as “The aim of our project is to investigate the immunologic parameters after HBV vaccination in mice modeling NAFLD, employing both standard and high-dose vaccination protocols. Following vaccination, we conducted comprehensive analyses, which include antibody titers, inflammatory and anti-inflammatory T and B lymphocyte ratio in peripheral blood and associated tissues where the lymphocyte maturation are occured and/or completed, to provide a detailed understanding of the immune response.”(Page 2, LÄ°ne 77-80)
Comments 4: More phishing than design unless more clearly explained to the reader. Some attempt is done in the discussion but I cannot easily detect a clear logic from aim of the study towards conclusions.
Response 4: Discussion section was revised.
In the discussion section, we first focused on the changes observed in immunological parameters, which are critical for understanding the immune response in NAFLD. We then presented the histopathological changes, such as hepatic steatosis, lobular inflammation, and hepatocellular ballooning, to provide a comprehensive view of the liver pathology. This sequence was chosen to build a logical progression from the underlying immune responses to the resulting tissue changes. “The observation of high hepatocyte ballooning scores and in NAFLD mice following high-dose vaccination suggests that even safe levels of vaccine adjuvants may have toxic effects in cases of steatosis. This finding leads to the conclusion that double-dose vaccination may not be effective or safe in NAFLD cases.” Was added to conclusion section.
Comments 5: Help the reader, certainly the less educated immunologist to better understand.
Response 5: It is expected that less educated immunologists will improve their knowledge on the subject by consulting reference books.
Comments 6: It is interesting that the ethics committee approved the protocol although from the introduction it remains initially unclear why animals were needed from data that more meaningful could have been obtained from humans. The authors did various more detailed investigations that look - as outlined - more like an add on than a clear conceptual design as outlined and therefore the animals needed to be submitted to experiments
Response 6: Due to our study design, we needed homogeneous NAFLD cases without HBV vaccination. Since it was not possible to create the desired experimental groups with human subjects under the conditions of our country having national HBV vaccination, the research was carried out with animal models. Additionally, by using the C57BL-6 inbred mouse line, we could minimize individual variation due to high genetic homogeneity. However, individual variations are much higher in human subjects. Therefore, animal experiments are used to create specific, homogeneous conditions that are difficult to achieve in humans. Based on this evidence, our research involving animal experiments was approved by the Hacettepe University Animal Experiments Ethics Committee.
In the methodology section, the study design, including the nutritional model, vaccination protocol, and immunological parameters, is described in separate subsections.
Comments 6re: It is my impression that the authors use often immunity where I suggest they use immune response and in some situations immunogenicity would appear to be a more appropriate term
Response 6re: Agree. Thank you for your valuable comments. We have revised the term 'Immunity' to 'immune response' or 'immunogenicity' as appropriate.
Comments 7: Vaccination for HBV is for 2 reasons important in patients with chronic liver disease (see for example review Joshi et al, 2021 etc.) 1. Excessive mortality and morbidity in case of (any) underlying liver disease. (Therefore also HAV vaccination important). 2. Prevention of HBV and the spectrum of related pathology including risk HCC.
Response 7: Introduction was revised as recomendation.
“Both non-alcoholic fatty liver disease (NAFLD) and chronic hepatitis B infection are common causes of chronic liver diseases around the world, with global prevalenc-es of about 25.2% and 3.9%, respectively [1-3]. NAFLD progresses from the initial stage of simple fatty liver, characterized by reversible fat accumulation, to the more severe stage of non-alcoholic steatohepatitis (NASH), marked by active tissue damage. The progression of NASH can lead to liver cirrhosis, and potentially hepatocellular carcinoma. NAFLD is commonly associated with metabolic syndromes such as obesity, insulin resistance, type 2 diabetes, and dyslipidemia, and is linked to an increased overall mortality compared to the general population [4].” Was added to introduction section.
“Hepatitis B virus (HBV) infection significantly increases the risk of death due to cirrhosis and hepatocellular carcinoma (HCC). Although global infection rates have fallen in recent years thanks to effective vaccines and extensive vaccination efforts, 5-10% of individuals fail to produce an antibody response, limiting the vaccine's over-all effectiveness [19-20].” Was present in introduction section.
“Studies have found that obese individuals are at a higher risk of infection complications and tend to have lower vaccine immunity compared to those with a healthy weight. Specifically, the response to hepatitis B virus (HBV) vaccines is often diminished in obese individuals [12-14]. Nonetheless, recombinant HBV vaccines are still considered effective and safe for this population [15-17].” Was added to introduction section.
Agreed, as mentioned in the introduction section: 'Chronic hepatitis B infection is a common cause of chronic liver diseases... Hepatitis B virus (HBV) infection significantly increases the risk of death due to cirrhosis and hepatocellular carcinoma (HCC). Although global infection rates have fallen in recent years thanks to effective vaccines and extensive vaccination efforts, 5-10% of individuals fail to produce an antibody response, limiting the vaccine's overall effectiveness.' The introduction provides brief information on the necessity of HBV vaccination. However, the focus of the paper is on vaccine efficacy in the NAFLD group, as this is the primary objective of the study.
Comments 8: It remains somewhat puzzling why the authors study in detail certain aspects of the immune response but do not describe any further detail of pathology. It is hard to see (image quality?) if, for example, any inflammatory response or fiborisis in the NAFLD livers has occurred. Only H and E stains?(difficult to see in (b) specimen: steatohepatitis? The response various tremendously among humans. Therefore the "pure storage of fat" (steatosis) is different from significant inflammation (steatohepatitis)
Response 8: Agree. Thank you very much for bringing this shortcoming to our attention and giving us the opportunity to expand the scope of our article.
Metod section was revised as “In order to determine the occurrence of NAFLD in liver samples taken; We used the NAFLD activity score (NAS), which was designed and validated by the Pathology Committee of the NASH Clinical Research Network. The proposed NAS is the unweighted sum of steatosis, lobular inflammation, and hepatocellular ballooning scores [26, 27]. Liver steatosis scored as 0 (<5% steatosis), 1 (6-33% steatosis), 2 (34-66% steatosis), 3 (67-100% steatosis); lobular inflammation as 0 (no foci), 1 (<2 foci per 200X field), 2 (2-4 foci per 200X field), and 3 (>4 foci per 200X field); hepatocyte ballooning as 0 (none), 1 (few balloon cells), 2 (many cells/prominent ballooning). Liver samples were also examined for fibrosis (0-4) [26, 27]” (Page 4, LÄ°ne 163-170).
We have revised Figure 2 (page 7), better quality new images were captured, and scale bars were added. Additionally, new images (Figure 3) have been added to show ballooning degeneration of hepatocytes and lobular inflammation in the livers (page 8).
Results section was revised as “At the end of the in-vivo experiments, the levels of hepatic steatosis (Figure 2), lobular inflammation, and hepatocellular ballooning (Figure 3) in liver tissues from all animals were determined through histopathological examination and NAS was calculated. It was observed that 81% of the mice in the NAFLD group exhibited hepatic steatosis, while only one animal in the Control group showed mild steatosis. The mean steatosis scores in the NAFLD groups were 0.86 [median (1st-3rd quartile): 1 (0-2)] for the NAFLD/ND group and 1.79 [2(1-3)] for the NAFLD/HD2-HD3 group (p=0.067). Within the NAFLD group, four mice had third-degree steatosis, six had second-degree steatosis, and seven had first-degree steatosis; in four mice, no hepatic steatosis was detected. Statistical analysis indicated a significant difference in hepatic steatosis between the Control and NAFLD groups (p<0.001). The mean [median (1st-3rd quartile)] lobular inflammation scores were 1.5 [1.5 (1-2)] in the Control/ND group, 1.58 [2 (1-2)] in the Control/HD2-HD3 group, 1.29 [1 (1-2)] in the NAFLD/ND group, and 1.64 [2 (1-2)] in the NAFLD/HD2-HD3 group. There was no significant difference in lobular inflammation scores between the groups (p=0.809 for the Control group, p=0.525 for the NAFLD group). Hepatocyte ballooning was not detected in any mice in the Control group. However, the hepatocyte ballooning scores in the NAFLD/HD2-HD3 group were significantly higher than in the NAFLD/ND group [mean [median (1st-3rd quartile)]: 1.43 [2 (1-2)] vs. 0.71 [1 (0-1)]; p=0.046]. The mean [median (1st-3rd quartile)] NAS scores were 1.50 [1.5 (1-2)] in the Control/ND group and 1.67 [2 (1-2)] in the Control/HD2-HD3 group (p=0.616). However, the scores were 2.86 [3 (1-5)] in the NAFLD/ND group and 4.86 [5 (3.75-6.25)] in the NAFLD/HD2-HD3 subgroups (p=0.056). In the histopathological evaluation, no fibrosis was detected in any liver samples.” (Page 6, Line 253-277).
The abstract, discussion, and consclusion sections were also revised in line with the new histopathological findings we added.
Comments 9: Abstract, do we need to know specifics of diet in abstract?
Response 9: Agree. Diet details were removed from the abstract (page 1, line 19)
Comments 10: Introduction can be shortened and focus more on presumed pathobiology patients with advanced liver disease tend to be poorer vaccine responders in general than healthy people also patients with chronic diseases
Response 10: Agree. Introduction section was revised and focused more on presumed pathobiology patients with advanced liver disease; “NAFLD progresses from the initial stage of simple fatty liver, characterized by reversible fat accumulation, to the more severe stage of non-alcoholic steatohepatitis (NASH), marked by active tissue damage. The progression of NASH can lead to liver cirrhosis, and potentially hepatocellular carcinoma. NAFLD is commonly associated with metabolic syndromes such as obesity, insulin resistance, type 2 diabetes, and dyslipidemia, and is linked to an increased overall mortality compared to the general population [4].”. “The liver is a primary innate immune organ, densely populated with various in-nate immune cells, including natural killer (NK) cells, and Kupffer cells (KCs). These cells is shown to play crucial roles in the excessive production of hepatic Th1 cytokines in NAFLD [7].” was added (Page 1 and Page 2).
Comments 11: the authors rightly note the unpredictability of responses due to many variables: are their groups big enough for detailed analysis?
Response 11: Generally, groups of 6-10 animals are commonly used in laboratory animal studies. This number can provide sufficient data for a study's effect size and statistical power goals, as long as homogeneous groups are created. In addition, C57BL-6 mice, an inbred mouse strain, were used in our study to keep genetic variation between subjects to a minimum.
“Harrison, R. D. (2017). "Sample Size Determination in Preclinical Animal Studies: A Review." Frontiers in Veterinary Science, 4, 231.” This review article discusses methods for determining sample sizes in animal studies and provides guidance on the number of animals required for reliable results.
However, “When the NAS values were compared among groups with 6-7 mice, borderline significance was observed. Increasing the sample size in future studies may reveal a significant difference in this parameter.” Was added to the limitation section of manuscript (Page 16, Line 516-521).
Comments 12: The high risk groups are?
Response 12: “Investigating vaccination in high-risk groups remains crucial, and while numerous studies have explored hepatitis B vaccination in obese individuals from risk groups, a notable gap exists in research specifically focusing on hepatitis B vaccination in individuals with NAFLD.” Was revised as “While some studies have examined hepatitis B vaccination in obese individuals [13,14,18,20] there remains a significant gap in research specifically addressing hepati-tis B vaccination in individuals with NAFLD [18].”
Comments 13: M and M: is all detail necessary (like determination of antibody titers etc etc.)?
Response 13: We affirm that all methodological details are essential. These details have been deemed necessary and sufficient by other reviewers.
Comments 14: Minor: 3500 kcal etc is the total calories fed to the animals during the study period if correctly understood
Response 14:
Misunderstood. To clarify, the calorie values provided refer to the energy content per kilogram of diet (page 3, lines 112 and 114).
The control diet contained 3500 kilocalories per kilogram, while the HFD/CD diet, prepared to induce NAFLD, contained 4650 kilocalories per kilogram. During the in vivo experiments, mice on the control diet consumed an average of 2.5-3 g of feed per day, while those on the HFD/CD diet consumed an average of 2-2.5 g per day. As a result, the daily energy intake for mice in the control group was approximately 9.6 kilocalories per mouse, whereas mice in the NAFLD group, fed with the HFD/CD diet and fructose solution, consumed approximately 14.9 kilocalories per mouse daily

Reviewer 4 Report
Comments and Suggestions for Authors
I've been invited to provide the peer-review of the paper "Investigation of Hepatitis-B Vaccine Immunity in Non-Alcoholic Fatty Liver Disease Mouse Model" from the study group led by Kütük T. In this study, 36 mice were managed in an experimental model in order to ascertain whether HBV immunity induced by vaccination is (or not) lower among NAFLD-affected mice compared to healthy ones; a high-dose HBV vaccine program affects or not vaccinated mice.
According to the results hereby reported, mice affected by experimental NAFLD showed lower CD27+ cell count and higher CD38+ cell count, with no remarkable differences after HBV vaccination, but vaccination levels were associated with a different Treg level.
In other words, while the paper suggests that only limited differences between NAFLD and control groups could be identified in terms of vaccine efficacy, several specificities could be identified in terms of interplay between immune system cells.
From my point of view, the present paper could be accepted for publication after the fixing of several issues, and more precisely:
1) The English text is mostly appropriate, but particularly abstract and several sections avross the main text are affected by annoying typos that must be fixed (e.g. "Than HbsAb titers, T follicular helper, T follicular regulatory, CD27+, CD38+ cells in the blood were analysed and immunofluorescent stainings in sections of the thymus and spleen were done" ... did you mean "The HBsAb titers, etc). Please check the spelling of main acronyms across the main text, particularly HBsAg, HBsAb etc.
2) Statistical analysis must be improved. According to Table 1 and Table 3, you did perform repeated 1 vs 1 comparisons of several variables through Mann-Whitney test. Also Kruskal-Wallis test has been performed. However, while K-W test copes with the repeated comparisons, M-W does not. By repeating 1 vs. 1 comparisons, alpha error similarly increases. Therefore, please report the results of multiple comparisons directly by K-W test, avoiding repeated M-W tests. On the contrary, subsequent analyses by means of GLM are appropriate and do not require further refinements.
Comments on the Quality of English LanguageThe English text is mostly appropriate, but particularly abstract and several sections avross the main text are affected by annoying typos that must be fixed (e.g. "Than HbsAb titers, T follicular helper, T follicular regulatory, CD27+, CD38+ cells in the blood were analysed and immunofluorescent stainings in sections of the thymus and spleen were done" ... did you mean "The HBsAb titers, etc). Please check the spelling of main acronyms across the main text, particularly HBsAg, HBsAb etc.
Author Response
Comments 1: I've been invited to provide the peer-review of the paper "Investigation of Hepatitis-B Vaccine Immunity in Non-Alcoholic Fatty Liver Disease Mouse Model" from the study group led by Kütük T. In this study, 36 mice were managed in an experimental model in order to ascertain whether HBV immunity induced by vaccination is (or not) lower among NAFLD-affected mice compared to healthy ones; a high-dose HBV vaccine program affects or not vaccinated mice. According to the results hereby reported, mice affected by experimental NAFLD showed lower CD27+ cell count and higher CD38+ cell count, with no remarkable differences after HBV vaccination, but vaccination levels were associated with a different Treg level. In other words, while the paper suggests that only limited differences between NAFLD and control groups could be identified in terms of vaccine efficacy, several specificities could be identified in terms of interplay between immune system cells.
From my point of view, the present paper could be accepted for publication after the fixing of several issues, and more precisely:
Response 1: Thank you for your positive feedback
Comments 2: The English text is mostly appropriate, but particularly abstract and several sections across the main text are affected by annoying typos that must be fixed (e.g. "Than HbsAb titers, T follicular helper, T follicular regulatory, CD27+, CD38+ cells in the blood were analysed and immunofluorescent stainings in sections of the thymus and spleen were done" ... did you mean "The HBsAb titers, etc). Please check the spelling of main acronyms across the main text, particularly HBsAg, HBsAb etc
Response 2: Agree. Thank you for your comment; To avoid confusion between Ag and Ab, throughout the article, 'HbsAb titers' were corrected to 'anti-HBs titers'.
In the abstract, the sentence “Than HbsAb titers, T follicular helper, T follicular regulatory, CD27+, CD38+ cells in the blood were analysed and immunofluorescent stainings in sections of the thymus and spleen were done” was corrected to "All mice were euthanized, blood samples for anti-HBs titers, T follicular helper, T follicular regulatory, CD27+, CD38+ cells, and liver, spleen and thymus for histopatologic evaluation were taken"
Additionally, the accuracy of the acronym 'HBsAg' and 'anti-HBs' was checked throughout the manuscript.
Comments 3: Statistical analysis must be improved. According to Table 1 and Table 3, you did perform repeated 1 vs 1 comparisons of several variables through Mann-Whitney test. Also Kruskal-Wallis test has been performed. However, while K-W test copes with the repeated comparisons, M-W does not. By repeating 1 vs. 1 comparisons, alpha error similarly increases. Therefore, please report the results of multiple comparisons directly by K-W test, avoiding repeated M-W tests. On the contrary, subsequent analyses by means of GLM are appropriate and do not require further refinements.
Response 3: Thank you for your attention and for allowing us to correct our mistake.
In the study, we wanted to demonstrate both single and multiple comparisons simultaneously. Statistical analysis in Method section was improved as “The variation in variables from prevaccination to postvaccination was analyzed with the Wilcoxon Signed Ranks Test. Differences in variables between two independent groups (Control and NAFLD groups) were examined using the Mann-Whitney U test. Differences in variables among three vaccination groups (ND, HD2, HD3) were examined using the Kruskal-Wallis test.”.
'Wilcoxon Signed Ranks Test' was added to Foodnote of Table 1.
Comments 4: Comments on the Quality of English Language: The English text is mostly appropriate, but particularly abstract and several sections avross the main text are affected by annoying typos that must be fixed (e.g. "Than HbsAb titers, T follicular helper, T follicular regulatory, CD27+, CD38+ cells in the blood were analysed and immunofluorescent stainings in sections of the thymus and spleen were done" ... did you mean "The HBsAb titers, etc). Please check the spelling of main acronyms across the main text, particularly HBsAg, HBsAb etc.
Response 4: Agree. “HbsAb titers”were corrected as “anti-HBs titers”
Additionally, the accuracy of the acronym 'HBsAg' and 'anti-HBs' was checked throughout the manuscript.

Round 2
Reviewer 1 Report
Comments and Suggestions for Authors
The authors have addressed the noted observations. A more polished and precise work is observed in this new version of the manuscript. The figures have been significantly improved, and the text content has generally been enriched.
Author Response
Comments: The authors have addressed the noted observations. A more polished and precise work is observed in this new version of the manuscript. The figures have been significantly improved, and the text content has generally been enriched.
Response: We would like to sincerely thank you for your constructive feedback on our manuscript. Your insightful comments have greatly contributed to enhancing the quality of our work. The improvements in the figures and the enriched text content are direct results of the valuable suggestions you provided. It is gratifying to know that the revised version is seen as a more polished and precise piece of work. Your expertise and time spent reviewing our manuscript are truly appreciated.
Reviewer 2 Report
Comments and Suggestions for Authors
The authors have sufficiently addressed my comments.
Author Response
Comments: The authors have sufficiently addressed my comments.
Response: We greatly appreciate your feedback and are glad to hear that we have sufficiently addressed your comments. Your constructive suggestions have been instrumental in improving the quality of our manuscript, and we are grateful for the time and expertise you invested in reviewing our work.
Reviewer 3 Report
Comments and Suggestions for Authors
I regret to conclude that I am still unconvinced that this article should be published. It is as if the authors come from another world re the conceptual design and impact of NASH etc. I asked for a pathobiological concept for their immunological observations and I am asked to consult the literature. It is as if we are now the mentors and supervisors of this article.
Author Response
Comments 1: I regret to conclude that I am still unconvinced that this article should be published.
Response 1: We believe that the revisions we made based on your constructive feedback have strengthened our manuscript. It is disappointing, however, that you still feel unsatisfied.
Comments 2: It is as if the authors come from another world regarding the conceptual design and impact of NASH, etc.
Response 2: Our study was conducted by a multidisciplinary team, including an experienced pediatrician responsible for the vaccination studies master's program, a veterinarian responsible for laboratory animals and serving as an animal ethics committee consultant, a pathologist, and an immunologist. Our research design was focused on modeling NAFLD in mice, not on modeling NASH. However, you repeatedly suggest that we do not understand the conceptual framework and impact of NASH.
The conceptual design and clinical output of the study are presented in Lines 79-91: (Conceptual Design of the Study) 'This study was designed to evaluate the immunological and histopathological responses to different vaccination protocols involving high and normal doses of an HBV vaccine in mice modeling NAFLD. The primary focus is on the comparison of HbsAb titers, T follicular helper cells (Tfh), T follicular regulatory cells (Tfr), and specific immune cell populations such as CD27+ and CD38+ cells across the different dosing regimens. Additionally, the study includes a comprehensive histopathological evaluation of key organs, including the liver, spleen, and thymus, to assess any potential tissue changes or damage induced by the different vaccine doses.' (Clinical Input) 'By integrating both immunological and histopathological data, the study aims to provide a detailed understanding of the dose-dependent effects of the HBV vaccine on the immune system and tissue integrity in NAFLD cases. By comparing the outcomes across high and normal doses of HBV, this study will contribute valuable information on the safety and effectiveness of the vaccine, ultimately guiding clinical decision-making for optimal dosing strategies.'
The rationale for conducting this study on laboratory animals is explained in Lines 427-432: 'Due to the national HBV vaccination program in our country, it was not feasible to obtain homogeneous NAFLD cases without HBV vaccination from human subjects. Therefore, we conducted the research using animal models. By employing the C57BL-6 inbred mouse line, we minimized individual genetic variation, which is challenging to control in human subjects. Animal experiments allowed us to create specific, uniform conditions essential for our study that are difficult to achieve in humans.'
Comments 3: "I asked for a pathobiological concept for their immunological observations, and I am asked to consult the literature. It is as if we are now the mentors and supervisors of this article."
Response 3: In none of our responses to your comments in Round 1 did we suggest that you need to consult the literature. The response we provided was directed towards less experienced immunologists, not towards you personally. As seen in dialog; “R1_C5: Help the reader, certainly the less educated immunologist, to better understand.” “R1_R5: 'It is expected that less educated immunologists will improve their knowledge on the subject by consulting reference books.”As seen in dialog; “R1_C5: Help the reader, certainly the less educated immunologist, to better understand.” “R1_R5: 'It is expected that less educated immunologists will improve their knowledge on the subject by consulting reference books.”
Reviewer 4 Report
Comments and Suggestions for Authors
The paper has been improved and fixed according to my recommendations. No further actions needed.
Author Response
Comment: The paper has been improved and fixed according to my recommendations. No further actions needed.
Response: We would like to express our sincere gratitude for your valuable feedback. Your recommendations have played a crucial role in improving and refining our paper. We are pleased to hear that the revisions meet your expectations and that no further actions are needed.